# Direct and Indirect Detoxification Effects of Humic Substances

**Lydia Bondareva** [1,*] **and Nadezhda Kudryasheva** [2,3]

1   Federal Scientific Center of Hygiene Named after F.F. Erisman, 141014 Moscow, Russia
2   Federal Research Center 'Krasnoyarsk Scientific Center Siberian Branch Russian Academy of Sciences', Institute of Biophysics, 660036 Krasnoyarsk, Russia; n-qdr@yandex.ru
3   Biophysics Department, Siberian Federal University, 660041 Krasnoyarsk, Russia
*   Correspondence: lydiabondareva@gmail.com; Tel.: +7-495-586-12-76

**Abstract:** The review summarizes studies on the detoxification effects of water-soluble humic substances (HS), which are products of the natural transformation of organic substances in soils and bottom sediments that serve as natural detoxifying agents in water solutions. The detoxifying effects of HS on microorganisms are quite complex: HS neutralize free pollutants (indirect bioeffects) and also stimulate the protective response of organisms (direct bioeffects). Prospects and potential problems of bioluminescent bacteria-based assay to monitor toxicity of solutions in the presence of HS are discussed. The main criterion for the bioassay application is versatility and ease of use. The detoxification efficiency of HS in different pollutant solutions was evaluated, and the detoxification mechanisms are discussed. Particular attention was paid to the direct and complex direct + indirect effects of HS. The review focuses on the protective function of HS in solutions of radionuclides and salts of stable metals, with special consideration of the antioxidant properties of HS.

**Keywords:** humic substances; detoxification; luminous bacteria; adaptive response





## 1. Introduction

Humic substances (HS) are complex mixtures of high-molecular organic compounds of natural origin. HS are formed as a result of the decomposition of plant and animal residues under the influence of microorganisms and abiotic environmental factors [1]. HS are found in soil, rivers and lakes [2], and important HS matrices include sediment, peat, coal and solid fossil fuels. Their content in soil and water is 60–80% of the total organic matter and in peat and coal it ranges from 20 to 90% [3].

The composition, structure and properties of HS are strongly affected by their age and degree of humification. The generally accepted classification of HS [3] is based on the difference in solubility in acids and alkalis. According to this classification, HS are divided into three components: humin—non-recoverable residue, insoluble in either alkalis or acids; humic acids—HS fraction, soluble in alkalis and insoluble in acids (at pH < 2); and fulvic acids—HS fraction, soluble in both alkalis and acids. The term "humic substances" is used as a general name for both humic and fulvic acids. Humic acids are the most mobile and reactive component of HS and they are actively involved to chemical and biochemical processes in ecosystems.

The most important functional groups of HS are the carboxylic, phenolic, alcoholic, carbonyl, amino and sulfhydryl groups; the predominance of the carboxyl groups contributes to the acidic character of HS [3–6]. The identified functional groups of HS are also involved in the structure of biomolecules, including sugars, fatty acids, polypeptides and amino acids.

Previously, HS were considered as inert compounds, especially in relation to living organisms. The main topics discussed were the physical and chemical characteristics of HS, such as its acid status, light absorption, and photolytic reactions that release low-molecular-weight HS fragments. The influence of HS on soil and water inhabitants have

attracted interest in the last few decades. It is known that the detoxifying effects of HS on microorganisms are quite complex. HS do not only neutralize free pollutants in solutions (indirect bioeffects), but may also stimulate the protective response of organisms (direct bioeffects). These effects are a topic of interest due to the multiple interrelations between HS and microorganisms including the humification of organic matter in soils and sediments.

There are few articles in the literature on the indirect effects of HS on organisms, and even fewer papers on the direct effects of HS. However, the role of HS in both aquatic and terrestrial ecosystems is extensive. In the long term, dissolved HS may determine or significantly alter the chemistry of entire reservoirs and act as critical natural agents to remove pollutants from natural water. In the medium term, HS may act as a source of nutrients and rapidly change the concentration and toxicity of organic and inorganic pollutants. In the short term, HS may act as natural xenobiotics, which produce various biological effects and thus may influence water communities; the different mechanisms involved in these changes are also under consideration [7,8].

HS are known to produce both toxic and stimulative effects on microorganisms. The value of HS LD-50 is 0.536 g per kilogram [9], which confirms the harmlessness of high-concentration solutions of HS. On other hand, microbial activity increases in soils with a higher content of HS up to 300 ppm [9].

This review elucidates the main signaling events that govern one of the important functions of HS, that is the detoxification of pollutants in the aquatic environment in order to shed more light on the nature, properties, dynamics and functions of HS as part of the ecosystem. We pay particular attention to the direct and complex direct + indirect effects of HS, and endogenous and exogenous redox transformations are taken into consideration. Special attention is paid to the protective function of HS in the solutions of radionuclides and salts of stable metals.

## 2. Influence of HS on Living Organisms

Numerous studies have shown that HS enhance root, leaf and shoot growth and also stimulate the germination of various crop species [10–14]. These positive effects are explained by the involvement of HS in various physiological and metabolic processes [11,12,15]. The addition of HS stimulates nutrient uptake, [16] cell permeability [17], and seems to regulate the mechanisms for stimulating plant growth [11,14,18–21].

It is not easy to differentiate the direct and indirect effects of HS [22]. some of the positive effects are ascribed to the general improvement of soil fertility and higher nutrient availability in plants, while in other cases, HS seem to positively influence the metabolic and signaling pathways of plant development by acting directly on specific physiological targets [11,14]. Therefore, understanding the biological activity of HS and the molecular mechanisms through which they exert their functions is an important ecological task and a valid tool in facing environmental problems.

The beneficial actions of HS in the environment has been ascribed to its two main complementary effects on aquatic systems [5,23]: (1) the HS effects are a consequence of their previous actions on the properties of water solutions. These effects are known as indirect effects and mainly result from the ability of HS to form stable natural chelates or complexes with metals in soil [3]; and (2) the HS effects that result from the direct interaction of HS with the cell membranes of living organisms. The latter are known as direct effects and affect the growth of living organisms through a complex network of signaling pathways regulated by the main plant hormones and effectors, such as auxin, nitric oxide, ethylene, abscisic acid, cytokinin and reactive oxygen species [21].

Several other factors also influence the intensity of direct and indirect effects of HS on living organisms. These factors can be classified as intrinsic (related to the intrinsic physicochemical properties of HS such as structure-conformation, size distribution, etc.) and extrinsic (related to crop management, the presence of abiotic or biotic stresses, etc.) [22,24].

The review of Trevisan et al. [25] analyzed the main signal events governing the effects of HS on plant metabolism and shed more light on the nature, properties, dynamics and

functions of HS as a part of soil and agricultural ecosystems. For example, photochemical degradation of dissolved organic matter can play an important role in carbon cycling in natural water, either directly by the photochemical production of volatile carbon species or indirectly through the production of $CO_2$ by sequential photochemical/biological oxidation, etc. [25].

The main effects associated with the impact of HS on the environment according to [2,25] are shown in Table 1.

**Table 1.** Environmental issues involving humic substances (HS) [25].

| Issue | Role of Humic Substances |
| --- | --- |
| Carbon cycling | Major C pool, transformations, transport and accumulation |
| Light penetration into waters | Absorption and attenuation of light by humic chromophores |
| Soil warming | Absorption of solar radiation by soil humic matter |
| Soil and water acidification | Binding of protons, aluminium and base cations in soils and water |
| Nutrient source | Reservoir of carbon, nitrogen, phosphorous, sulphur and chlorine |
| Nutrient control | Binding of iron and phosphate |
| Microbial metabolism | Substrate for microbes |
| Weathering | Enhancement of mineral dissolution rate |
| Soil formation (podzolisation) | Translocation of dissolved humic substances and associated metals (Al, Fe) |
| Properties of fine sediments | Adsorption at surfaces and alteration of colloidal properties |
| Soil structure | Aggregation effect on soil minerals solids |
| Photochemistry | Mediation of light-driven reactions |
| Heavy metals | Binding, transport, influence on bioavailability, redox reactions |
| Pesticides, xenobiotics | Binding, transport, influence on bioavailability |
| Radioactive waste disposal | Binding and transport of radionuclide ions in groundwaters |
| Ecosystem buffering | Control of proton and metal ions concentrations, persistence |
| Direct biological effect | Uptake and direct interaction with living organisms |

The indirect effects of HS in ecosystems had been discussed extensively and have focused on the indirect effects of HS on organisms including heavy metals and nutrient control, as well as the modulation of the toxicity of pesticides and other xenobiotics. However, in the broader sense, all the issues mentioned in Table 1 (except the last item) represent a variety of possible indirect effects of HS on living organisms.

A specific mechanism of HS action in plants was shown by Pflugmacher et al. [26,27]. They revealed that environmentally relevant concentrations of HS can induce modulation of photosynthetic oxygen release in alga *Scenedesmus armatus*, water moss *Vesicularia dubyana* and hornwort *Ceratophyllum demersum* [7,26,27].

The influence of HS on sodium metabolism in *Daphnia magna* was investigated by Glover et al. [28]. Environmentally relevant levels of Suwannee River natural organic matter and commercial HS significantly enhanced the sodium influx, which was characterized by an increased maximal sodium transport rate and uptake affinity. In the subsequent study, two types of the above-mentioned HS were compared and a mechanism of action was proposed [29]. At pH 4, HS promoted a linear sodium uptake kinetic relationship, which was attributed to the altered membrane permeability due to enhanced membrane binding of HS at low pH. In contrast, a natural organic matter elicited no consistent action on sodium influx. These results suggest that impacts on sodium metabolism may be limited to certain types of HS [29].

Previously, HS were generally considered to be bioinert and were used exclusively as sorbents for metals and organic pollutants [30,31], thus producing indirect effects on

organisms. However, some papers have shown that HS may act as chemical xenobiotics. Meinelt et al. [32] found that synthetic HS (HS1500) affects the physiological state and sex ratio of the fish *Xiphophorus helleri*. In addition, the hormone-like effect of HS on the nematode *Caenorhabditis elegans* was established in [7,33], and an increase in the number of amphipod deaths and changes in biochemical parameters was also shown [34] The toxicity of aquatic solutions was found to be dependent on the concentration and redox potentials of quinones in studies by Kudryasheva [35,36] and Vetrova et al. [37] using bacterial bioassay, thus providing information on the bioeffects of quinones in the presence of HS.

### 3. Antioxidant Properties of HS via Bacterial Bioassay

Redox processes are a part of the main vital metabolic cycles, such as respiration and photosynthesis, and the excess of exogenous redox compounds (quinones, phenols, and multivalent metals) in the environment may disturb the redox equilibrium, resulting in the toxic impact of pollutants on organisms [38,39].

Quinones, being organic oxidizers, produce semiquinone radicals and other reactive oxygen species that have a harmful impact on water systems and their inhabitants. In nature, quinones can be produced as a result of the oxidative transformation of phenols, an abundant group of hydroxylated aromatic compounds that are known to be one of the top three most common pollutants (after metal salts and oil products). Phenols are frequent components in the wastewater of chemical recovery, organic synthesis, as well as hydrolytic, cellulose, and flax industries [40–42]. A range of phenolic substances are also synthesized and extracellularly excreted by a variety of soil bacteria and they are used as molecular signals in microbial communication and as adaptogens. They are prone to redox transformations in soils and aquifers, especially at low pH in the presence of Fe(III).

Currently, metal salts, including radionuclides, are a leading cause of environmental pollution; therefore, they deserve special attention. In studies by Tarasova et al. [43], the patterns of detoxification of solutions of inorganic pollutants were investigated using HS as a detoxifying agent. Complex salt $K_3(Fe(CN)_6)$ was chosen as a model oxidizer because of its stability in aqueous solution (as opposed to uncoordinated iron) and the monoelectronic oxidation $Fe^{3+}/Fe^{2+}$ transition [44].

In nature, the process of redox Fe transformation requires sufficient energy to overcome chemical barriers, and the efficiency of the transformation depends on temperature, pressure, pH, and Fe concentrations under environmental conditions. HS can chelate ferrous iron to form the Fe(II)-HA complex, which keeps Fe atoms at an effective distance from each other.

The detoxifying properties of HS in oxidant solutions are usually explained by their reducing ability [45,46]: phenolic, SH−, and other groups of HS macromolecules are responsible for the reduction of oxidants. The evidence that has accumulated shows that HS, and especially their quinoid fragments, may play an important role as electron carriers in microbial redox reactions involved in the biodegradation of pollutants [3].

The detoxifying ability of HS was also discussed in papers by Tarasova et al. [43,47,48]. It was shown that the mechanism of HS detoxification is complex and it includes: (1) chemical and physiochemical processes in solutions with binding and reduction of the oxidizers involved; and (2) intensification of the protective responses of organisms based on the increase in the rate of endogenous biochemical reactions, as well as stabilization and enhancement of mucus layers outside the cell walls.

The main feature of all classical biological tests is the integral response; this means that the action of all toxic compounds in solutions is complex and leads to changes in certain physiological functions. In this regard, it is impossible to determine the cause of the toxic effect (i.e., the type and concentration of toxic compounds) using only biological analyses. Another feature of biological tests is the non-additivity of the effects of numerous environmental pollutants and natural components. This feature means that the combined toxic effect of the sum of the compounds in the complex environmental solutions may be

greater or less than the sum of the individual effects of these compounds. This makes it impossible to assess the toxicity of a complex matrix based only on chemical data analysis. As a result, the current approach to environmental research requires a combination of chemical and biological methods. In addition, it should always be noted that chemical test-systems are initially calibrated using a standard biological system under specific standard environmental conditions.

The different sensitivity of biological assays based on different organisms is determined by the physiological, cellular and biochemical characteristics of the organisms; this should always be taken into account in the process of environmental research. It is assumed [49–51] that the combination of chemical and biological methods developed for environmental monitoring should include a set of biological analyses that differ in their sensitivity to pollutants. However, the sensitivity of biological tests should first be evaluated for individual substances and their combinations with each other [52].

The bioluminescence of marine bacteria is sensitive to toxic compounds; therefore, marine bacteria have been widely used for several decades as a bioassay in environmental toxicity assessment [51–56].

Bioassay systems that include luminous marine bacteria are important for the biotechnological application of the bioluminescence phenomenon [54–58]. The physiological parameter here is luminescence intensity, which can be easily measured instrumentally. The suppression of bioluminescence intensity evaluates the toxicity of the water solution. Bacterial bioluminescence assays can be based on biological systems of varying complexity and include bacteria or enzymes that are used to study the effect of toxic compounds on bacterial cells and enzymes, respectively [54,59–63]. Both bacterial and enzymatic assays have been successfully used to monitor toxicity, as well as antioxidant and pro-oxidant activity of carbon and Fe-containing nanoparticles (fullerene derivatives and magnetite-based composites) [64–66]. One study [67] compared the antioxidant ability of fullerenols and HS as macrostructures of artificial and natural origin, respectively. Both antioxidants demonstrated detoxifying effects at low-concentration, however, the detoxification efficiency of fullerenol was higher. The HS demonstrated moderate detoxification ability and its dependence on time, probably due to diffusion restrictions in the solutions of the HS macromolecules.

The first description of a biological test based on bioluminescent bacteria appeared in 1969, in the work by Kossler as reported by Grabert and Kossler [68]. In the late 1980s, the test was standardized in Germany as a method for detecting pollutants, and was later adapted for specific research [47,51–66].

Previous reviews [33,34] have proposed the classification of the effects of exogenous compounds on the bacterial bioluminescent system of enzymatic reactions. The classification of the effects of exogenous compounds considers the following mechanisms: (1) the effect of exogenous compounds on the population of electronically excited states of the bioluminescence emitter as a result of intermolecular energy transfer to exogenous acceptor molecules; (2) the change in the efficiency of intramolecular singlet-triplet energy migration in the presence of heavy halogen atoms; (3) the change in the rates of coupled reactions due to competition in the process of oxidation of the endogenous reducer, NADH, i.e., the process of hydrogen transfer (H = $e^-$ + $H^+$); (4) the interaction with enzymes and changes in their activity; and (5) the non-specific influence of exogenous donors and acceptors of electrons on the electronic density distribution in the bioluminescent system. Mechanisms 1 and 2 can be conditionally classified as "physical", 3 and 4 as "biochemical" and 5 as "integral physiochemical". Luminescent bacteria and enzymatic systems were used to study the detoxifying properties of HS [43,47,48], that is, the products of the natural transformation of organic substances in soil and bottom sediments. It has been shown in several papers [43,47,48] that HS can reduce the toxic effects of metal salts and organic oxidants, and also, under certain conditions, increase them. It has been demonstrated that HS not only reduce the content of toxic substances in the environment by binding and neutralizing them, but also interfere with metabolic processes, thus accelerating endogenous

reactions in the bioluminescent system, which demonstrates the "indirect" and "direct" effects of HS on microorganisms. These studies used an approach based on the variation of exogenous compounds with a focus on their physico-chemical characteristics and primary physico-chemical processes in the simplest bioassay system. This approach suggests a fixed detoxifying agent; however, it is supposed that further studies could vary HS preparations to have different properties. Several studies [43,47,48] have used Gumat-80 preparation ("Gumat", Irkutsk, Russia) as a source of HS. It is produced by the non-extracting treatment of coal with alkali. The characteristics of the preparation are: humic acids ≈ 85%, soluble potassium—9%, iron—1%, water—5%, pH 8–9 in 1% water solution.

The most significant illustration of the detoxification effect of HS in radioactive solutions is presented in [68]. Luminous bacteria *P. phosphoreum* 1883 IBSO from the Collection (CCIBSO 836) of the Institute of Biophysics SB RAS, Russia was used as a bioassay; solutions of anthropogenic radionuclide Am-241 were detoxified. The solutions of $Am(NO_3)_3$ with a concentration of $10^{-10}$ M in 1.1 M $NaNO_3$ (pH 7.0) and activity of 3,0 Bq·L$^{-1}$ were applied as a source of ionizing radiation. The radioactivity of the solutions was measured using gamma spectrometry and liquid-scintillation spectrometry. The "RadSpectraDec" ("Radon", Moscow, Russian Federation) program package was applied in the latter case. The Gumat-80 preparation was used as a source of HS. The HS concentration of 0.25 g·L$^{-1}$, which did not change the bioluminescent intensity, was chosen for the experiments. To evaluate the effect of Am-241 on the bacteria, the relative bioluminescent intensity $I^{rel}$ was calculated as:

$$I^{\mathrm{rel}} = \frac{I_1}{I_{contr}} \tag{1}$$

where, $I_1$ is bacterial bioluminescence intensity in a radioactive solution; $I_{contr}$ is the bacterial bioluminescence intensity in a control (nonradioactive) solution.

Values of $I^{rel}$ were plotted vs. time of exposure to Am-241. The $I^{rel}$ in the absence of HS was compared to that in their presence.

Figure 1 shows the schematic dependence of bioluminescence intensity on exposure time in the control (nonradioactive) and radioactive solutions. As can be seen, the radiation exposure suppressed the glow of bacteria (Curve 1) and presence of HS mitigated the suppression by moving the kinetic curve closer to the control (Curve 2).

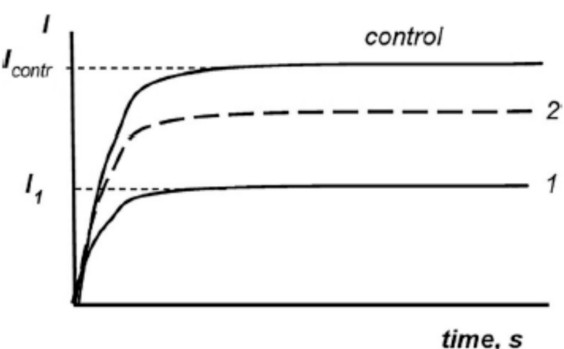

**Figure 1.** Principle of toxicity measurements: dependence of bioluminescent intensity (*I*) on time in a control sample (control) in the presence of a toxic compound (1) and in a solution of a toxic compound + detoxifying agent, HS (2).

The migration behavior of toxic and radiotoxic metal ions, e.g., actinides, strongly depends on their oxidation state, which can be influenced by HS [69]. A detailed description of the influence of HS on the mobility of actinides in the environment requires an understanding of the effects of HS on the oxidation states of actinides, as well as knowledge about the actinide ion complexation by HS.

For example, Bio-Rex70 has been used as a reference substance to study the nature of metal complexation sites in HS. Bio-Rex70 is a cation exchange resin that, in contrast to HS, has no phenolic OH groups but exclusively carboxyl groups that are capable of binding

the metal ions. Thus, Th(IV), Np(IV) and Np(V) sorbates on Bio-Rex70 were studied to determine the structural parameters for the interaction of the actinide ions with carboxyl groups [70]. In the case of Np(V), the influence of HS phenolic/acidic OH groups on the interaction between Np(V) and HS was studied at pH 7, applying modified HS with blocked phenolic/acidic OH groups in addition to the unmodified HS.

In order to study the redox properties of humic acids and the redox stability of actinide humate complexes in detail, Choppin [71] developed model synthetic humic acids with pronounced redox functionalities.

Figure 2a shows the relationship between the relative bioluminescence intensity ($I^{rel}$) and the time of exposure to Am-241 in the absence (Curve 1) and presence (Curve 2) of HS [72]. It can be seen that Am-241 did not affect the intensity of bioluminescence for the first 4 h of exposure in both systems (Curves 1 and 2, Figure 2a). However, further exposure (up to 20 h) led to the activation of bioluminescence ($I^{rel} > 1$). Longer exposure to the radionuclide (>20 h) resulted in the inhibition of bioluminescence ($I^{rel} < 1$).

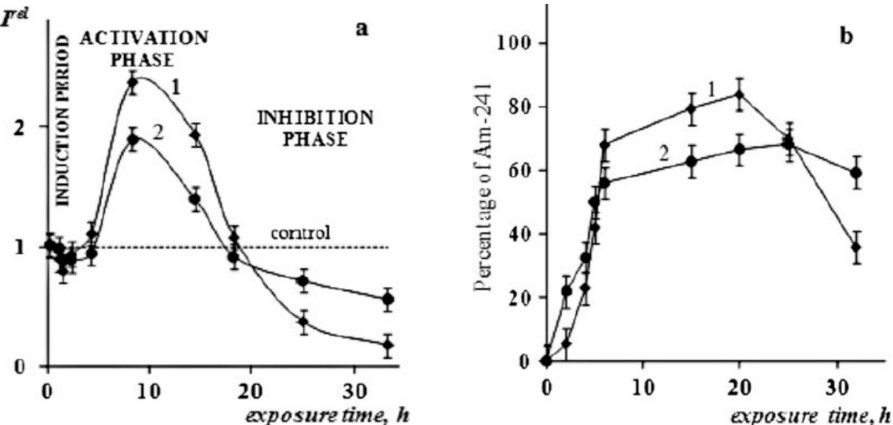

**Figure 2.** Chronic effects of Am-241 on luminous bacteria: (**a**) relative bioluminescence intensity ($I^{rel}$) vs. time of exposure, (**b**) accumulation of Am-241 in the cellular fraction. 1—absence of HS, 2—presence of HS. The activity of the Am-421 solution was 3 kBq·L$^{-1}$ (C = 10$^{-10}$ M) and the concentration of HS was 0.25 g·L$^{-1}$ [69].

Hence, Figure 2a shows that HS decreased the intensity of the bioluminescence during the activation period and increased it during the inhibition period, thereby mitigating the difference between the control (nonradioactive) and radioactive samples. It was concluded that HS reduce (mitigate) the effects of Am-241 radioactivity. Figure 2b shows that the presence of HS in the bacterial suspension decreases accumulation of Am-241 in the bacterial cells, excluding the final time period of the observation. The latter effect was explained as the destruction of the bacterial cells by radioactivity of Am-241 [69]. The authors attributed these experimental results to binding of the radionuclide into a complex with HS and the "masking effect" of HS in the water suspensions, i.e., the indirect effect of HS on the microorganism. The results demonstrate the detoxifying properties of HS in the radioactive solutions of alpha-emitting radionuclide Am-241.

The actual reality of Am-241 in nature-like environments needs to be attended to. This radionuclide is a by-product of the decay of radioactive weapon plutonium, and is characterized by its long lifetime (432.6 years); it is currently accumulated in the environment and promises to be a real environmental problem for future generations.

Electron microscopy has been used to search for evidence of the "masking effect" of HS in the bacterial suspension with Am-241. The images are shown in Figure 3, which shows the control cells (Figure 3a) and bacterial cells fixed with radionuclide at the inhibition and activation stage in the absence and presence of HS (Figure 3b–e).

The control sample cells had a regular oval shape with a smooth cell envelope and included small electron-transparent fragments. The control cell shape did not change during

the entire experiment. In the case of chronic exposure to Am-241, the visible ultrastructure of the cells showed damage during the activation (Figure 3b) and the inhibition period (Figure 3c). Many pleomorphic cells and slit cells were observed on ultrathin sections of the samples. There were some differences in the bacteria ultrastructure in the activation (Figure 3b) and inhibition (Figure 3c) periods. There were many normal cells during the activation period, however, all cell was damaged during the inhibition period.

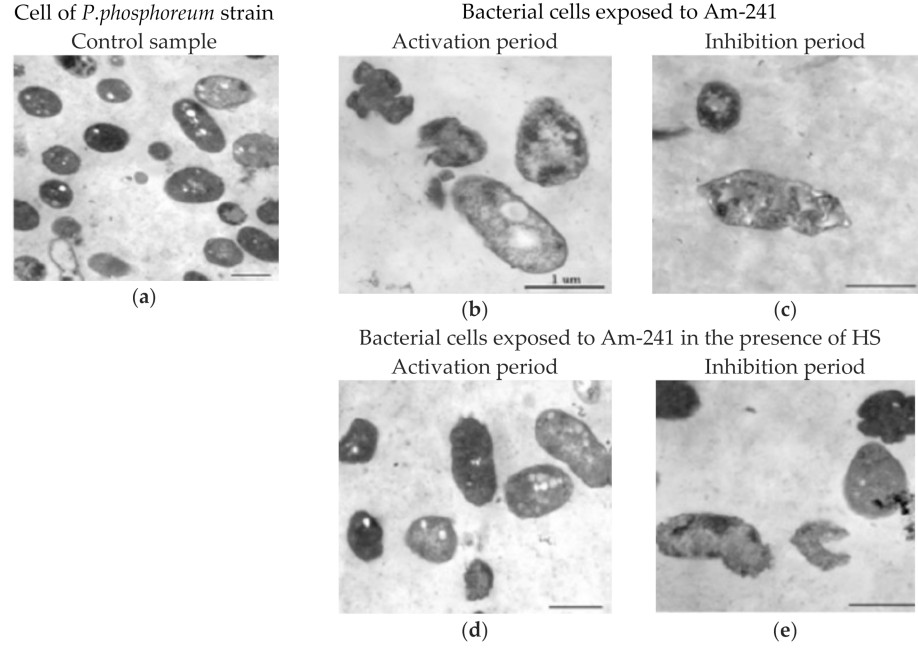

**Figure 3.** Ultrastructure of *Photobacterium phosphoreum* exposed to Am-241 in the absence (**b**,**c**) and in the presence (**d**,**e**) of HS: (**a**) control samples, (**b**,**d**) activation period, (**c**,**e**) inhibition period. Activity of Am-241 solution—3kBq·L$^{-1}$ (C = 10$^{-10}$ M). Concentration of HS—0.25 g·L$^{-1}$ [69].

Figure 3d,e show the cellular changes in Am-241 solutions in the presence of HS. Damaged cells were also present. However, during the activation period, there were less damaged cells. The inhibition period showed the presence of intact cells (Figure 3d,e), which was in contrast to the system without HS (Figure 3b,c).

Less damage of cells in the presence of HS (Figure 4) may also be associated with the decrease in the accumulation of Am-241 in the cell fraction (Figure 3b), probably due to the steric effect of a large complex of HS and Am-241 ion.

In [72], the combined effects of beta-emitting radionuclide tritium and HS on luminous bacteria were studied. The choice of tritium was justified by its environmental prevalence; it is one of the most common decay products in the nuclear industry. Tritium is able to activate or inhibit the bioluminescence of marine bacteria and their enzymes [61,73,74]. The effects of tritium on bacteria suspensions are presented in Figure 4 a–c, Curve 1. The HS were shown to decrease the inhibition and activation effects of tritium (Figure 4 a,b), similar to those of Am-241 [69]. The HS did not change the bioluminescence intensity in the absence of the tritium effect (Figure 4c). Correlations between the bioluminescence intensity and the content of reactive oxygen species (ROS) were found in the radioactive bacterial suspensions in [72], hence, the involvement of ROS in the detoxification processes was proved.

Electron microscopy has also been used to study the effect of HS on bacterial cells in solutions of non-radioactive toxic compounds [47,48,75].

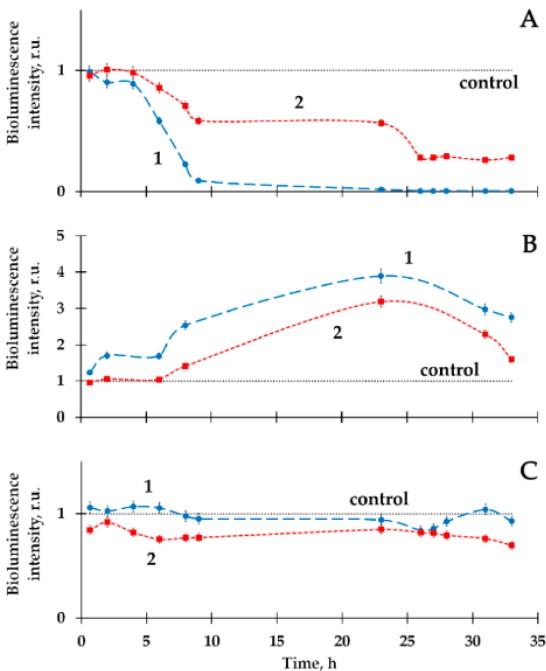

**Figure 4.** Bacterial bioluminescence kinetics in tritiated water in the absence (1) and presence (2) of HS. Specific radioactivity: (**A**) 2 MBq·L$^{-1}$; (**B**) 50 MBq·L$^{-1}$; (**C**) 200 MBq·L$^{-1}$. HS concentration—10$^{-3}$ g·L$^{-1}$ [72].

Figure 5 shows microscopic images for solutions of two model toxic compounds: CrCl$_3$ and tetrafluoro-1,4-benzoquinone [47,48]. It was found that, in the presence of HS, specific changes appeared in the cell membranes. In most cells, an amorphous substance with an average electron density was present outside of the cell walls (see arrows in Figure 5). It seems to represent the remains of a mucous layer fixed by HS macromolecules. It is important that the remains of a mucous capsule were not observed in toxic solutions without HS or in the solutions of HS without toxicants. Mucous capsules are known [76] to protect bacteria from antimicrobial agents. They are almost always present on the surface of cells growing in nature (as opposed to on laboratory cultures). While preparing samples for the electron microscopy procedure, the mucous layers are usually partly washed away.

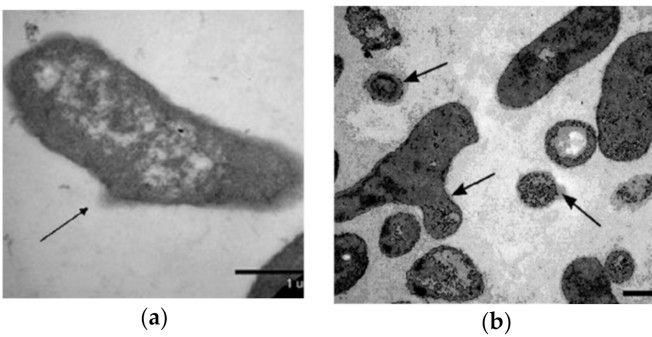

(**a**)　　　　　　　　　　　　(**b**)

**Figure 5.** Ultrastructure of *P. phosphoreum* batch culture affected by: (**a**) CrCl$^{3+}$ + HS [42] and (**b**) tetrafluoro-1,4-benzoquinone + HS. Arrows denote fragments of mucous capsules. *Bar* = 1 μm [59].

The bacteria were thought [50] to intensify the synthesis of the extracellular matrix as a response to the unfavorable influence of toxic compounds (CrCl$_3$ or benzoquinone). Most likely, the HS macromolecules enhance and stabilize the mucous capsule, thus intensifying the protective response of the cells. These results show the direct and probably complex +indirect effects of HS.

## 4. Conclusions

Humic substances (HS) are ubiquitous natural products of the decomposition of dead organic matter. For a long time, HS have been generally considered to be inert in ecosystems. Thus, the ecotoxicological aspects of HS have mainly been discussed in connection with their ability to bind heavy metals and various organic xenobiotics, with consequent modulation of their bioavailability and toxicity. These types of effects on organisms are called indirect effects [77]

To identify HS–organism interactions, priority should be given to understanding the functional mechanisms that have the highest impact on abiotic stress protection. HS interact with biochemical components and affect the signaling pathways, eliciting dynamic signaling crosstalk inside the microorganisms in the course of various types of stresses involving the toxic impact of anthropogenic pollutants [78,79].

HS do function as natural detoxifying agents and are able to mitigate the bioeffects of anthropogenic pollutants, including radionuclides. We suggest that luminous bacteria-based bioassay is the most promising bioassay system for monitoring the processes occurring in natural ecosystems. The ability of HS to mitigate the effects of alpha-emitting radionuclide Am-241 and beta-emitting radionuclide tritium on the luminous bacteria has been analyzed in this review. Toxicological bioluminescence monitoring and microscopic studies of the effects of radionuclides on the bacterial cells in the presence and absences of HS were discussed. It was demonstrated that HS serve as a protective shield for aquatic microorganisms exposed to radionuclides.

The choice of radionuclides was justified by the potential risks of their impact on the environment in the next decades and centuries. Am-241 is a by-product of the radioactive decay of weapon plutonium, which is characterized by its long decay lifetime (432.6 years), and it is currently accumulated in the environment. Tritium is a product of a lot of radiochemical processes and one of the most common decay products in the nuclear industry; it accumulates around nuclear plants and after nuclear incidents.

The ability of HS to protect the microorganisms in solutions of organic and inorganic pollutants has also been demonstrated.

Evidence for the direct effects of HS on bacteria were presented, such as an increase in the rate of NADH-dependent enzymatic processes in cells and the synthesis of mucous layers fixed by HS in the toxic solutions at cell membranes. Hence, the direct bioeffects of HS in toxic solutions might form the basis for the stimulation of protective bacterial responses. Thus, the process of detoxification of anthropogenic pollutants with HS was evaluated and discussed.

**Author Contributions:** Conceptualization, L.B. and N.K.; methodology, L.B.; validation, N.K.; formal analysis L.B.; investigation, N.K.; resources, L.B., N.K.; data curation, N.K.; writing—original draft preparation, L.B.; writing—review and editing, N.K.; visualization, L.B. and N.K.; supervision, N.K.; project administration, L.B.; funding acquisition, L.B. All authors have read and agreed to the published version of the manuscript.

**Funding:** This research received no external funding.

**Institutional Review Board Statement:** Not applicable.

**Informed Consent Statement:** Not applicable.

**Data Availability Statement:** Data sharing not aplicable.

**Acknowledgments:** This review was prepared with the partial financial support of the Program of the Federal Service for Surveillance on Consumer Rights Protection and Human Wellbeing (Russian Federation) 2020–2025.

**Conflicts of Interest:** The authors declare no conflict of interest.

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
