# Peer review of "Direct and Indirect Detoxification Effects of Humic Substances"

_agronomy, doi:10.3390/agronomy11020198_

Round 1
Reviewer 1 Report
The authors present the study of the detoxification effects of the interaction humic substances with living organisms.
There are very interesting subject, but the manuscript the is outside the scope of the Agronomy journal and its special issue.
The manuscript is not well organized. The article would be more readable and understandable if it included such sections as: Material and Methods as well as Result or/i Discussion. Part include Discussion is very short and contains only 5 literature cited. Beside the section Introduction should be re-written.
Additionally, papier does not explain, why Gumat-80 preparation was selected to experiment and why it was decided to test its properties in an aquatic environment, since there is a soil remediation preparation. How a source of natural humic substances is not a good example. It's not clear. In my opinion papier should contains chemical analysis of this preparation.
Author Response
There are very interesting subject, but the manuscript the is outside the scope of the Agronomy journal and its special issue.
Reply: We revised the manuscript substantially, paying a special attention to the role of water-soluble humic substances in soil processes, namely, detoxification ability of HS with special attention to radioactive contaminations (alpha and beta radionuclides) and non-radioactive (organic and inorganic) ones. The changes and additions are marked with colors. Particular attention was paid to the differentiation of direct and indirect effects HS on microorganisms.
The manuscript is not well organized. The article would be more readable and understandable if it included such sections as: Material and Methods as well as Result or/i Discussion. Part include Discussion is very short and contains only 5 literature cited. Beside the section Introduction should be re-written.
Reply: We reorganized the manuscript, the concept of the manuscript was clarified and detalized, the title of the manuscript was changed. We suggested three new sections of the review. We did not use sections Material and Methods as well as Result or/and Discussion as we suppose that their specificity is not in accordance with a Review but only with the Research paper.
Additionally, paper does not explain, why Gumat-80 preparation was selected to experiment and why it was decided to test its properties in an aquatic environment, since there is a soil remediation preparation. How a source of natural humic substances is not a good example. It's not clear. In my opinion papier should contains chemical analysis of this preparation.
Reply: we introduced the following paragraph (marked with blue color in the text of paper, lines 237-234): These studies used the approach based on variations of exogenous compounds with a focus on their phisico-chemical characteristics and primary physico-chemical processes in the simplest bioassay system. This approach suggests a fixed detoxifying agent; however, it is supposed that further studies can vary HS preparations with different properties. The papers [46, 50-51] used Gumat-80 preparation (“Gumat”, Irkutsk, Russia) as a source of HS. It was produced by non-extracting treatment of coal with alkali. Characteristics of the preparation are: humic acids ≈ 85%, soluble potassium – 9%, iron – 1%, water – 5%, pH 8-9 in 1% water solution.
Reviewer 2 Report
The considered review article surveys the contemporary knowledge on studying the biological effects of humic substances (HS), natural detoxifying agents, products of oxidative transformation of organic matter in soils and bottom sediments. It is noteworthy that bacteria-based bioluminescence assay constitutes an efficient technique for examining the detoxifying ability of HS, and the authors outline and discuss its advantages. The authors specify the two detoxification modes described in the literature which are based on the two groups of detoxification mechanisms: (i) indirect mechanisms, which refer to the chemical processes in water solutions responsible for the observed effects, and (ii) direct mechanisms which pertain to the influence of HS (or HS combination with toxic compounds) on organisms which result in protective response of water microorganisms. The second mode merits the most attention. The material presented in the manuscript is unquestionably timely and certainly of interest for a general readership and, most prominently, for agronomy and ecology experts. I strongly recommend this work for publication in Agronomy. The minor suggestions of technical character, which need to be addressed prior to publication, follow below:
- I would suggest to revise a bit Conclusion section. Firstly, I propose to draw therein a distinction between the two above-mentioned detoxification modes and, secondly, I feel that the last two paragraphs need to be rephrased to make them more concise and clear.
- I suggest to alter the title of the article. "To" should not appear ahead of the title phrase. An appropriate title, which conveys the manuscript content, would be the following: "Direct and indirect detoxification effects of humic substances".
- Since the abbreviation "HS" is introduced, it should be used throughout the whole text of the article.
- On page 1 (line 26), it should be "dark-colored" instead of "dark-coloured".
- I would rephrase the paragraph below Table 1 (page 3, lines 84-88) to make its main idea more clear.
- I suggest to check the references in terms of their style for their compliance with the Journal format and correct them if necessary.
Author Response
We are thankful for the very useful remarks.
- I would suggest to revise a bit Conclusion section. Firstly, I propose to draw therein a distinction between the two above-mentioned detoxification modes and, secondly, I feel that the last two paragraphs need to be rephrased to make them more concise and clear.
Reply: We revised Conclusion section. Additional information (with two modes involved) is marked with yellow.
- I suggest to alter the title of the article. "To" should not appear ahead of the title phrase. An appropriate title, which conveys the manuscript content, would be the following: "Direct and indirect detoxification effects of humic substances".
Reply: Thank you for this very useful suggestion, we changed the title and reconstructed some parts of the manuscript with special attention to direct and indirect detoxification effects. Herewith, we corrected a concept of our presentation.
- Since the abbreviation "HS" is introduced, it should be used throughout the whole text of the article.
Reply: corrected
- On page 1 (line 26), it should be "dark-colored" instead of "dark-coloured".
Reply: corrected
- I would rephrase the paragraph below Table 1 (page 3, lines 84-88) to make its main idea more clear.
Reply: The rephrased variant follows below: “Until recently, mainly indirect effects of HS in ecosystems had been discussed. Indirect effects of HS on organisms include both heavy metals and nutrients control, as well as the modulating of the toxicity of pesticides and other xenobiotics. However, in the broader sense, all the issues mentioned in Table 1 (except the last item) represent a variety of possible indirect effects of HS on living organisms.”
- I suggest to check the references in terms of their style for their compliance with the Journal format and correct them if necessary.
Reply: We checked the list of references carefully.
Reviewer 3 Report
1) line 27:
I do not agree that humic substances are resistant to degradation. It is well known that they are formed by degradation of plants etc. and their composition, structure a and properties are strongly affected by their age and degree of humification.
2) line 29:
not only in soils, rivers and lakes – important matrices are e.g. sediments, peats and (coals lignite and leonardite)
3) line 40:
humus acids or humic acids?
4) lines 45-55:
see 1)Living organisms are integral to soil organic matter and participate in its functioning (including humification).
5) line 216:
Equations must be numbered.
6) Figure 1:
Is the figure prepared by authors? or adopted (in this case, the source reference is necessary)?
7) Figure 2:
dtto - see 6)
8) Figure 3:
dtto - see 6)
9) It is not clear, what is the aim of this paper. Conclusions presented in this manuscript are well known and too general, e.g. “...HS have unique properties and characteristics...” and “ ... dissolved HS have to be considered abiotic ecological driving forces, somewhat less obvious than temperature, nutrients, or light...”
Author Response
We are thankful for the very useful remarks.
1) line 27:
I do not agree that humic substances are resistant to degradation. It is well known that they are formed by degradation of plants etc. and their composition, structure a and properties are strongly affected by their age and degree of humification.
Reply: We changed the sentence to the following one: “Humic substances (HS) are complex mixtures of high-molecular organic compounds of natural origin. HS are formed as a result of decomposition of plant and animal residues under influence of microorganisms and abiotic environmental factors [1]”. Additionally, we added the sentence “HS’ composition, structure and properties are strongly affected by their age and degree of humification.” at the beginning of the following paragraph.
2) line 29:
not only in soils, rivers and lakes – important matrices are e.g. sediments, peats and (coals lignite and leonardite)
Reply: The reconstructed sentence: “HS are found in soils, rivers and lakes [2]; important HS matrices are sediments, peats, coals and solid fossil fuels .”
3) line 40:
humus acids or humic acids?
Reply: Corrected for HS
4) lines 45-55:
see 1) Living organisms are integral to soil organic matter and participate in its functioning (including humification).
Reply: We discuss integral responses of bacteria (which are used as bioassay systems) to the pollutants and HS in the process of bio-monitoring (lines 191-204 and further)
5) line 216:
Equations must be numbered.
Reply: Done
6) Figure 1-5:
Is the figure prepared by authors? or adopted (in this case, the source reference is necessary)?
Reply: The sources were presented for Figs 2-5.
9) It is not clear, what is the aim of this paper. Conclusions presented in this manuscript are well known and too general, e.g. “...HS have unique properties and characteristics...” and “ ... dissolved HS have to be considered abiotic ecological driving forces, somewhat less obvious than temperature, nutrients, or light...”
Reply: We removed these phrases.
Round 2
Reviewer 1 Report
Organic matter affects the accumulation of pollutants, but at the same time is a source of essential nutrients and energy for organisms participating in the processes of pollutant decomposition. These impacts depend largely on the properties of the pollutants. Exogenous organic matter may contribute to the intensification or decresing of the processes between organic matter and pollutants. That's way the subject of this paper are very interesting.
After major revision, the paper became legible and understandable and now fulfils the special issue scope of the Agronomy.
The current manuscript organisation is acceptable.
Author Response
Thank you very much for the review report and for your opinion.
Reviewer 3 Report
ad 1) line 27:
OK
ad 2) line 29:
OK
ad 33) line 40:
humus acids or humic acids?
Reply: Corrected for HS
Abbreviation HS means humic substances. Humic acids (or humus acids) must be defined better.
ad 4) lines 45-55:
see 1) Living organisms are integral to soil organic matter and participate in its functioning (including humification).
Reply: We discuss integral responses of bacteria (which are used as bioassay systems) to the pollutants and HS in the process of bio-monitoring (lines 191-204 and further)
The text (lines 45-55) must be modified.
ad 5) line 216:
OK
ad 6) Figure 1-5:
Is the figure prepared by authors? or adopted (in this case, the source reference is necessary)?
Reply: The sources were presented for Figs 2-5.
Sources must be preneted for all adopted figures.
ad 9) It is not clear, what is the aim of this paper. Conclusions presented in this manuscript are well known and too general, e.g. “...HS have unique properties and characteristics...” and “ ... dissolved HS have to be considered abiotic ecological driving forces, somewhat less obvious than temperature, nutrients, or light...”
Reply: We removed these phrases.
It is still not clear, what is the aim of this paper. Please, specify it in detail.
Author Response
ad 33) line 40:
humus acids or humic acids?
Reply: Corrected for HS
Abbreviation HS means humic substances. Humic acids (or humus acids) must be defined better.
Reply: We removed abbreviation HA. We mean “Humic acids” and do not use term “humus acids”
ad 4) lines 45-55:
see 1) Living organisms are integral to soil organic matter and participate in its functioning (including humification).
Reply: We discuss integral responses of bacteria (which are used as bioassay systems) to the pollutants and HS in the process of bio-monitoring (lines 191-204 and further)
The text (lines 45-55) must be modified.
Reply: It is assumed all over the text that the effect of HS is integral for all organisms with microorganisms involved. This is a common point of view now. We suppose that the first sentence in introduction is concerned with functions of microorganisms in soils: “Humic substances (HS) are complex mixtures of high-molecular organic compounds of natural origin. HS are formed as a result of decomposition of plant and animal residues under influence of microorganisms and abiotic environmental factors [1].” Additionally, a we mentioned that “These effects are of a topic interest due to multiple interrelations between HS and microorganisms including gumification of organic matter in soils and sediments” (lines 57-58)
ad 6) Figure 1-5:
Is the figure prepared by authors? or adopted (in this case, the source reference is necessary)?
Reply: The sources were presented for Figs 2-5.
Sources must be preneted for all adopted figures.
Reply: Sources for all adopted figures are presented (Figs 2-5). Figure 1 is modified by authors.
ad 9) It is not clear, what is the aim of this paper. Conclusions presented in this manuscript are well known and too general, e.g. “...HS have unique properties and characteristics...” and “ ... dissolved HS have to be considered abiotic ecological driving forces, somewhat less obvious than temperature, nutrients, or light...”
Reply: We removed these phrases.
It is still not clear, what is the aim of this paper. Please, specify it in detail.
Reply: The aim of the paper is presented at the end of the first section: “This review elucidates the main signaling events that govern one of the important functions of HS – detoxification of pollutants in the aquatic environment, in order to shed more light on the nature, properties, dynamics and functions of HS as part of the ecosystems. We pay a particular attention to the direct and complex direct+indirect effects of HS, with endogenous and exogenous redox transformations taken into consideration. Special attention is paid to the HS protective function in the solutions of radionuclides and salts of stable metals.”